# Worst-Case Regret Bounds for Exploration via Randomized Value Functions

**Daniel Russo**
Columbia University
djr2174@gsb.columbia.edu

## Abstract

This paper studies a recent proposal to use randomized value functions to drive exploration in reinforcement learning. These randomized value functions are generated by injecting random noise into the training data, making the approach compatible with many popular methods for estimating parameterized value functions. By providing a worst-case regret bound for tabular finite-horizon Markov decision processes, we show that planning with respect to these randomized value functions can induce provably efficient exploration.

## 1 Introduction

Exploration is one of the central challenges in reinforcement learning (RL). A large theoretical literature treats exploration in simple finite state and action MDPs, showing that it is possible to efficiently learn a near optimal policy through interaction alone [5, 8, 10, 11, 13–16, 25, 26]. Overwhelmingly, this literature focuses on optimistic algorithms, with most algorithms explicitly maintaining uncertainty sets that are likely to contain the true MDP.

It has been difficult to adapt these exploration algorithms to the more complex problems investigated in the applied RL literature. Most applied papers seem to generate exploration through $\epsilon$–greedy or Boltzmann exploration. Those simple methods are compatible with practical value function learning algorithms, which use parametric approximations to value functions to generalize across high dimensional state spaces. Unfortunately, such exploration algorithms can fail catastrophically in simple finite state MDPs [See e.g. 24]. This paper is inspired by the search for principled exploration algorithms that both (1) are compatible with practical function learning algorithms and (2) provide robust performance, at least when specialized to simple benchmarks like tabular MDPs.

Our focus will be on methods that generate exploration by planning with respect to randomized value function estimates. This idea was first proposed in a conference paper by [22] and is investigated more thoroughly in the journal paper [24]. It is inspired by work on posterior sampling for reinforcement learning (a.k.a Thompson sampling) [20, 27], which could be interpreted as sampling a value function from a posterior distribution and following the optimal policy under that value function for some extended period of time before resampling. A number of papers have subsequently investigated approaches that generate randomized value functions in complex reinforcement learning problems [6, 9, 12, 21, 23, 28, 29]. Our theory will focus on a specific approach of [22, 24], dubbed *randomized least squares value iteration* (RLSVI), as specialized to tabular MDPs. The name is a play on the classic least-squares policy iteration algorithm (LSPI) of [17]. RLSVI generates a randomized value function (essentially) by judiciously injecting Gaussian noise into the training data and then applying applying LSPI to this noisy dataset. One could naturally follow the same template while using other value learning algorithms in place of LSPI.

This is a strikingly simple algorithm, but providing rigorous theoretical guarantees has proved challenging. One challenge is that, despite the appealing conceptual connections, there are significant subtleties to any precise link between RLSVI and posterior sampling. The issue is that posterior

sampling based approaches are derived from a true Bayesian perspective in which one maintains beliefs over the underlying MDP. The approaches of [6, 9, 12, 23, 24, 28, 29] model only the value function, so Bayes rule is not even well defined.[1] The work of [22, 24] uses stochastic dominance arguments to relate the value function sampling distribution of RLSVI to a correct posterior in a Bayesian model where the true MDP is randomly drawn. This gives substantial insight, but the resulting analysis is not entirely satisfying as a robustness guarantee. It bounds regret on average over MDPs with transitions kernels drawn from a particular Dirichilet prior, but one may worry that hard reinforcement learning instances are extremely unlikely under this particular prior.

This paper develops a very different proof strategy and provides a worst-case regret bound for RLSVI applied to tabular finite-horizon MDPs. The crucial proof steps are to show that each randomized value function sampled by RLSVI has a significant probability of being optimistic (see Lemma 4) and then to show that from this property one can reduce regret analysis to concentration arguments pioneered by [13] (see Lemmas 6, 7). This approach is inspired by frequentist analysis of Thompson sampling for linear bandits [2] and especially the lucid description of [1]. However, applying these ideas in reinforcement learning appears to require novel analysis. The only prior extension of these proof techniques to tabular reinforcement learning was carried out by [3]. Reflecting the difficulty of such analyses, that paper does not provide regret bounds for a pure Thompson sampling algorithm; instead their algorithm samples many times from the posterior to form an optimistic model, as in the BOSS algorithm [4]. Also, unfortunately there is a significant error that paper's analysis and the correction has not yet been posted online, making a careful comparison difficult at this time.

The established regret bounds are not state of the art for tabular finite-horizon MDPs. Two steps in the proofs introduce extra factors of $\sqrt{S}$ in the bounds, where $S$ denotes the number of states. I hope some smart reader can improve this by intelligently adapting the techniques of [5, 11]. However, the primary goal of the paper is not to give the tightest possible regret bound, but to broaden the set of exploration approaches known to satisfy polynomial worst-case regret bounds. To this author, it is both fascinating and beautiful that carefully adding noise to the training data generates sophisticated exploration and proving this formally is worthwhile.

## 2   Problem formulation

We consider the problem of learning to optimize performance through repeated interactions with an unknown finite horizon MDP $M = (H, \mathcal{S}, \mathcal{A}, P, \mathcal{R}, s_1)$. The agent interacts with the environment across $K$ episodes. Each episode proceeds over $H$ periods, where for period $h \in \{1, \ldots, H\}$ of episode $k$ the agent is in state $s_h^k \in \mathcal{S} = \{1, \ldots, S\}$, takes action $a_h^k \in \mathcal{A} = \{1, \ldots, A\}$, observes the reward $r_h^k \in [0, 1]$ and, for $h < H$, also observes next state $s_{h+1}^k \in \mathcal{S}$. Let $\mathcal{H}_{k-1} = \{(s_h^i, a_h^i, r_h^i) : h = 1, \ldots H, i = 1, \ldots, k-1\}$ denote the history of interactions prior to episode $k$. The Markov transition kernel $P$ encodes the transition probabilities, with

$$P_{h,s_h^k,a_h^k}(s) = \mathbb{P}(s_{h+1}^k = s \mid a_h^k, s_h^k, \ldots, a_1^k, s_1^k, \mathcal{H}_{k-1}).$$

The reward distribution is encoded in $\mathcal{R}$, with

$$\mathcal{R}_{h,s_h^k,a_h^k}(dr) = \mathbb{P}\left(r_h^k = dr \mid a_h^k, s_h^k, \ldots, a_1^k, s_1^k, \mathcal{H}_{k-1}\right).$$

We usually instead refer to expected rewards encoded in a vector $R$ that satisfies $R_{h,s,a} = \mathbb{E}[r_{h,s,a}^k \mid s_h^k = s, a_h^k = a]$. We then refer to an MDP $(H, \mathcal{S}, \mathcal{A}, P, R, s_1)$, described in terms of its expected rewards rather than its reward distribution, as this is sufficient to determine the expected value accrued by any policy. The variable $s_1$ denotes a deterministic initial state and we assume $s_1^k = s_1$ for every episode $k$. At the expense of complicating some formulas, the entire paper could also be written assuming initial states are drawn from some distribution over $\mathcal{S}$, which is more standard in the literature.

A deterministic Markov policy $\pi = (\pi_1, \ldots, \pi_H)$ is a sequence of functions, where each $\pi_h : \mathcal{S} \to \mathcal{A}$ prescribes an action to play in each state. We let $\Pi$ denote the space of all such policies. We use $V_h^\pi \in \mathbb{R}^S$ to denote the value function associated with policy $\pi$ in the sub-episode consisting of

periods $\{h, \ldots, H\}$. To simplify many expressions, we set $V_{H+1}^\pi = 0 \in \mathbb{R}^S$. Then the value functions for $h \leq H$ are the unique solution to the the Bellman equations

$$V_h^\pi(s) = R_{h,s,\pi(s)} + \sum_{s' \in \mathcal{S}} P_{s,h,\pi(s)}(s') V_{h+1}^\pi(s') \qquad s \in \mathcal{S}, \; h = 1, \ldots, H.$$

The optimal value function is $V_h^*(s) = \max_{\pi \in \Pi} V_h^\pi(s)$.

An episodic reinforcement learning algorithm `Alg` is a possibly randomized procedure that associates each history with a policy to employ throughout the next episode. Formally, a randomized algorithm can depend on random seeds $\{\xi_k\}_{k \in \mathbb{N}}$ drawn independently of the past from some prespecified distribution. Such an episodic reinforcement learning algorithm selects a policy $\pi_k = \texttt{Alg}(\mathcal{H}_{k-1}, \xi_k)$ to be employed throughout episode $k$.

The cumulative expected regret incurred by `Alg` over $K$ episodes of interaction with the MDP $M$ is

$$\mathrm{Regret}(M, K, \texttt{Alg}) = \mathbb{E}_{\texttt{Alg}} \left[ \sum_{k=1}^K V_1^*(s_1^k) - V_1^{\pi_k}(s_1^k) \right]$$

where the expectation is taken over the random seeds used by a randomized algorithm and the randomness in the observed rewards and state transitions that influence the algorithm's chosen policies. This expression captures the algorithm's cumulative expected shortfall in performance relative to an omniscient benchmark, which knows and always employs the true optimal policy.

Of course, regret as formulated above depends on the MDP $M$ to which the algorithm is applied. Our goal is not to minimize regret under a particular MDP but to provide a guarantee that holds uniformly across a class of MDPs. This can be expressed more formally by considering a class $\mathcal{M}$ containing all MDPs with $S$ states, $A$ actions, $H$ periods, and rewards distributions bounded in $[0, 1]$. Our goal is to bound the worst-case regret $\sup_{M \in \mathcal{M}} \mathrm{Regret}(M, K, \texttt{Alg})$ incurred by an algorithm throughout $K$ episodes of interaction with an unknown MDP in this class. We aim for a bound on worst-case regret that scales sublinearly in $K$ and has some reasonable polynomial dependence in the size of state space, action space, and horizon. We won't explicitly maximize over $M$ in the analysis. Instead, we fix an arbitrary MDP $M$ and seek to bound regret in a way that does not depend on the particular transition probabilities or reward distributions under $M$.

It is worth remarking that, as formulated, our algorithm knows $S, A$, and $H$ but does not have knowledge of the number of episodes $K$. Indeed, we study a so-called *anytime* algorithm that has good performance for all sufficiently long sequences of interaction.

**Notation for empirical estimates.** We define $n_k(h, s, a) = \sum_{\ell=1}^{k-1} \mathbb{1}\{(s_h^\ell, a_h^\ell) = (s, a)\}$ to be the number of times action $a$ has been sampled in state $s$, period $h$. For every tuple $(h, s, a)$ with $n_k(h, s, a) > 0$, we define the empirical mean reward and empirical transition probabilities up to period $h$ by

$$\hat{R}_{h,s,a}^k = \frac{1}{n_k(h, s, a)} \sum_{\ell=1}^{k-1} \mathbb{1}\{(s_h^\ell, a_h^\ell) = (s, a)\} r_h^\ell \tag{1}$$

$$\hat{P}_{h,s,a}^k(s') = \frac{1}{n_k(h, s, a)} \sum_{\ell=1}^{k-1} \mathbb{1}\{(s_h^\ell, a_h^\ell, s_{h+1}^\ell) = (s, a, s')\} \quad \forall s' \in \mathcal{S}. \tag{2}$$

If $(h, s, a)$ was never sampled before episode $k$, we define $\hat{R}_{h,s,a}^k = 0$ and $\hat{P}_{h,s,a}^k = 0 \in \mathbb{R}^S$.

## 3 Randomized Least Squares Value Iteration

This section describes an algorithm called Randomized Least Squares Value Iteration (RLSVI). We describe RLSVI as specialized to a simple tabular problem *in a way that is most convenient for the subsequent theoretical analysis*. A mathematically equivalent definition – which defines RSLVI as estimating a value function on randomized training data – extends more gracefully . This interpretation is given at the end of the section and more carefully in [24].

At the start of episode $k$, the agent has observed a history of interactions $\mathcal{H}_{k-1}$. Based on this, it is natural to consider an estimated MDP $\hat{M}^k = (H, \mathcal{S}, \mathcal{A}, \hat{P}^k, \hat{R}^k, s_1)$ with empirical estimates of mean

rewards and transition probabilities. These are precisely defined in Equation (2) and the surrounding text. We could use backward recursion to solve for the optimal policy and value functions under the empirical MDP, but applying this policy would not generate exploration.

RLSVI builds on this idea, but to induce exploration it judiciously adds Gaussian noise before solving for an optimal policy. We can define RLSVI concisely as follows. In episode $k$ it samples a random vector with independent components $w^k \in \mathbb{R}^{HSA}$, where $w^k(h, s, a) \sim N\left(0, \sigma_k^2(h, s, a)\right)$. We define $\sigma_k(h, s, a) = \sqrt{\frac{\beta_k}{n_k(h,s,a)+1}}$, where $\beta_k$ is a tuning parameter and the denominator shrinks like the standard deviation of the average of $n_k(h, s, a)$ i.i.d samples. Given $w^k$, we construct a randomized perturbation of the empirical MDP $\overline{M}^k = (H, \mathcal{S}, \mathcal{A}, \hat{P}^k, \hat{R}^k + w^k, s_1)$ by adding the Gaussian noise to estimated rewards. RLSVI solves for the optimal policy $\pi^k$ under this MDP and applies it throughout the episode. This policy is, of course, greedy with respect to the (randomized) value functions under $\overline{M}^k$. The random noise $w^k$ in RLSVI should be large enough to dominate the error introduced by performing a noisy Bellman update using $\hat{P}^k$ and $\hat{R}^k$. We set $\beta_k = \tilde{O}(H^3)$ in the analysis, where functions of $H$ offer a coarse bound on quantities like the variance of an empirically estimated Bellman update. For $\beta = \{\beta_k\}_{k \in \mathbb{N}}$, we denote this algorithm by $\texttt{RLSVI}_\beta$.

**RLSVI as regression on perturbed data.** To extend beyond simple tabular problems, it is fruitful to view RLSVI–like in Algorithm 1–as an algorithm that performs recursive least squares estimation on the state-action value function. Randomization is injected into these value function estimates by perturbing observed rewards and by regularizing to a randomized prior sample. This prior sample is essential, as otherwise there would no be randomness in the estimated value function in initial periods. This procedure is the LSPI algorithm of [17] applied with noisy data and a tabular representation. The paper [24] includes many experiments with non-tabular representations. It should be stressed that although data-perturbations are sometimes used to regularize machine learning algorithms, here it is used only to drive exploration.

---

**Algorithm 1:** RLSVI for Tabular, Finite Horizon, MDPs

**input** : $H, S, A$, tuning parameters $\{\beta_k\}_{k \in \mathbb{N}}$

(1) **for** *episodes* $k = 1, 2, \ldots$ **do**
     /* Define squared temporal difference error                    */
(2)      $\mathcal{L}(Q \mid Q_{\text{next}}, \mathcal{D}) = \sum_{(s,a,r,s') \in \mathcal{D}} \left(Q(s, a) - r - \max_{a' \in \mathcal{A}} Q_{\text{next}}(s', a')\right)^2$ ;
(3)      $\mathcal{D}_h = \{(s_h^\ell, a_h^\ell, r_h^\ell, s_{h+1}^\ell) : \ell < k\} \qquad h < H$ ;                  /* Past data */
(4)      $\mathcal{D}_H = \{(s_H^\ell, a_H^\ell, r_H^\ell, \emptyset) : \ell < k\}$;

     /* Randomly perturb data                                          */
(5)      **for** *time periods* $h = 1, \ldots, H$ **do**
(6)          Sample array $\tilde{Q}_h \sim N(0, \beta_k I)$ ;                   /* Draw prior sample */
(7)          $\tilde{D}_h \leftarrow \{\}$;
(8)          **for** $(s, a, r, s') \in \mathcal{D}_h$ **do**
(9)              sample $w \sim N(0, \beta_k)$;
(10)              $\tilde{\mathcal{D}}_h \leftarrow \tilde{\mathcal{D}}_h \cup \{(s, a, r + w, s')\}$;
(11)          **end**
(12)      **end**

     /* Estimate $Q$ on noisy data                                 */
(13)      Define terminal value $Q_{H+1}^k(s, a) \leftarrow 0 \quad \forall s, a$ ;
(14)      **for** *time periods* $h = H, \ldots, 1$ **do**
(15)          $\hat{Q}_h \leftarrow \text{argmin}_{Q \in \mathbb{R}^{SA}} \mathcal{L}(Q \mid Q_{h+1}, \tilde{\mathcal{D}}_h) + \|Q - \tilde{Q}_h\|_2^2$ ;
(16)      **end**
(17)      Apply greedy policy with respect to $(\hat{Q}_1, \ldots \hat{Q}_H)$ throughout episode;
(18)      Observe data $s_1^k, a_1^k, r_1^k, \ldots s_H^k, a_H^k, r_H^k$ ;
(19) **end**

---

To understand this presentation of RLSVI, it is helpful to understand an equivalence between posterior sampling in a Bayesian linear model and fitting a regularized least squares estimate to

randomly perturbed data. We refer to [24] for a full discussion of this equivalence and review the scalar case here. Consider Bayes updating of a scalar parameter $\theta \sim N(0, \beta)$ based on noisy observations $Y = (y_1, \dots, y_n)$ where $y_i \mid \theta \sim N(0, \beta)$. The posterior distribution has the closed form $\theta \mid Y \sim N\left(\frac{1}{n+1}\sum_1^n y_i, \frac{\beta}{n+1}\right)$. We could generate a sample from this distribution by fitting a least squares estimate to noise perturbed data. Sample $W = (w_1, \dots, w_n)$ where each $w_i \sim N(0, \beta)$ is drawn independently and sample $\tilde{\theta} \sim N(0, \beta)$. Set $\tilde{y}_i = y_i + w_i$. Then

$$\hat{\theta} = \underset{\theta \in \mathbb{R}}{\operatorname{argmin}} \sum_{i=1}^{n} (\theta - \tilde{y}_i)^2 + (\theta - \tilde{\theta})^2 = \frac{1}{n+1}\left(\sum_{i=1}^{n}\tilde{y}_i + \tilde{\theta}\right) \tag{3}$$

satisfies $\hat{\theta} \mid Y \sim N\left(\frac{1}{n+1}\sum_1^n y_i, \frac{\beta}{n+1}\right)$. For more complex models, where exact posterior sampling is impossible, we may still hope estimation on randomly perturbed data generates samples that reflect uncertainty in a sensible way. As far as RLSVI is concerned, roughly the same calculation shows that in Algorithm 1 $\hat{Q}_h(s, a)$ is equal to an empirical Bellman update plus Gaussian noise:

$$\hat{Q}_h(s, a) \mid \hat{Q}_{h+1} \sim N\left(\hat{R}_{h,s,a} + \sum_{s' \in \mathcal{S}} \hat{P}_{h,s,a}(s') \max_{a' \in \mathcal{A}} \hat{Q}_{h+1}(s', a'), \frac{\beta_k}{n_k(h, s, a) + 1}\right).$$

It is worth noting that Algorithm 1 can be naturally applied to settings with function approximation. In line 15, instead of minimizing over all possible state-action value functions $Q \in \mathbb{R}^{S \times A}$, we minimize over the parameter $\theta$ defining some approximate value function $Q_\theta$. Instead of regularizing toward a random prior sample $\tilde{Q}_h$ in line 15, the methods in [24] regularize toward a random prior parameter $\tilde{\theta}_h$. See [23] for a study of these randomized prior samples in deep reinforcement learning.

## 4  Main result

Theorem 1 establishes that RLSVI satisfies a worst-case polynomial regret bound for tabular finite-horizon MDPs. It is worth contrasting RLSVI to $\epsilon$–greedy exploration and Boltzmann exploration, which are both widely used randomization approaches to exploration. Those simple methods explore by directly injecting randomness to the action chosen at each timestep. Unfortunately, they can fail catastrophically even on simple examples with a finite state space – requiring a time to learn that scales exponentially in the size of the state space. Instead, RLSVI generates randomization by training value functions with randomly perturbed rewards. Theorem 1 confirms that this approach generates a sophisticated form of exploration fundamentally different from $\epsilon$–greedy exploration and Boltzmann exploration. The notation $\tilde{O}$ ignores poly-logarithmic factors in $H, S, A$ and $K$.

**Theorem 1.** *Let $\mathcal{M}$ denote the set of MDPs with horizon $H$, $S$ states, $A$ actions, and rewards bounded in [0,1]. Then for a tuning parameter sequence $\beta = \{\beta_k\}_{k \in \mathbb{N}}$ with $\beta_k = \frac{1}{2}SH^3\log(2HSAk)$,*

$$\sup_{M \in \mathcal{M}} \operatorname{Regret}(M, K, \mathtt{RLSVI}_\beta) \leq \tilde{O}\left(H^3 S^{3/2}\sqrt{AK}\right).$$

This bound is not state of the art and that is not the main goal of this paper. I conjecture that the extra factor of $S$ can be removed from this bound through a careful analysis, making the dependence on $S$, $A$, and $K$, optimal. This conjecture is supported by numerical experiments and (informally) by a Bayesian regret analysis [24]. One extra $\sqrt{S}$ appears to come from a step at the very end of the proof in Lemma 7, where we bound a certain $L_1$ norm as in the analysis style of [13]. For optimistic algorithms, some recent work has avoided directly bounding that $L_1$-norm, yielding a tighter regret guarantee [5, 11]. Another factor of $\sqrt{S}$ stems from the choice of $\beta_k$, which is used in the proof of Lemma 5. This seems similar to an extra $\sqrt{d}$ factor that appears in worst-case regret upper bounds for Thompson sampling in $d$-dimensional linear bandit problems [1].

**Remark 1.** *Some translation is required to relate the dependence on $H$ with other literature. Many results are given in terms of the number of periods $T = KH$, which masks a factor of $H$. Also unlike e.g. [5], this paper treats time-inhomogenous transition kernels. In some sense agents must learn about $H$ extra state/action pairs. Roughly speaking then, our result exactly corresponds to what one would get by applying the UCRL2 analysis [13] to a time-inhomogenous finite-horizon problem.*

# 5 Proof of Theorem 1

The proof follows from several lemmas. Some are (possibly complex) technical adaptations of ideas present in many regret analyses. Lemmas 4 and 6 are the main discoveries that prompted this paper. Throughout we use the following notation: for any MDP $\tilde{M} = (H, \mathcal{S}, \mathcal{A}, \tilde{P}, \tilde{R}, s_1)$, let $V(\tilde{M}, \pi) \in \mathbb{R}$ denote the value function corresponding to policy $\pi$ from the initial state $s_1$. In this notation, for the true MDP $M$ we have $V(M, \pi) = V_1^\pi(s_1)$.

**A concentration inequality.** Through a careful application of Hoeffding's inequality, one can give a high probability bound on the error in applying a Bellman update to the (non-random) optimal value function $V_{h+1}^*$. Through this, and a union bound, Lemma bounds 2 bounds the expected number of times the empirically estimated MDP falls outside the confidence set

$$\mathcal{M}^k = \left\{ (H, \mathcal{S}, \mathcal{A}, P', R', s_1) : \quad \forall (h, s, a) | (R'_{h,s,a} - R_{h,s,a}) + \langle P'_{h,s,a} - P_{s,a,h}, V_{h+1}^* \rangle | \right.$$
$$\left. \leq \sqrt{e^k(h, s, a)} \right\}$$

where we define

$$\sqrt{e^k(h, s, a)} = H \sqrt{\frac{\log (2HSAk)}{n_k(s, h, a) + 1}}.$$

This set is a only a tool in the analysis and cannot be used by the agent since $V_{h+1}^*$ is unknown.

**Lemma 2** (Validity of confidence sets). $\sum_{k=1}^\infty \mathbb{P}\left( \hat{M}^k \notin \mathcal{M}^k \right) \leq \frac{\pi^2}{6}$.

**From value function error to on policy Bellman error.** For some fixed policy $\pi$, the next simple lemma expresses the gap between the value functions under two MDPs in terms of the differences between their Bellman operators. Results like this are critical to many analyses in the RL literature. Notice the asymmetric role of $\tilde{M}$ and $\overline{M}$. The value functions correspond to one MDP while the state trajectory is sampled in the other. We'll apply the lemma twice: once where $\tilde{M}$ is the true MDP and $\overline{M}$ is estimated one used by RLSVI and once where the role is reversed.

**Lemma 3.** *Consider any policy $\pi$ and two MDPs $\tilde{M} = (H, \mathcal{S}, \mathcal{A}, \tilde{P}, \tilde{R}, s_1)$ and $\overline{M} = (H, \mathcal{S}, \mathcal{A}, \overline{P}, \overline{R}, s_1)$. Let $\tilde{V}_h^\pi$ and $\overline{V}_h^\pi$ denote the respective value functions of $\pi$ under $\tilde{M}$ and $\overline{M}$. Then*

$$\overline{V}_1^\pi(s_1) - \tilde{V}_1^\pi(s_1) = \mathbb{E}_{\pi, \overline{M}} \left[ \sum_{h=1}^H \left( \overline{R}_{h, s_h, \pi(s_h)} - \tilde{R}_{h, s_h, \pi(s_h)} \right) + \langle \overline{P}_{h, s_h, \pi(s_h)} - \tilde{P}_{h, s_h, \pi(s_h)}, \tilde{V}_{h+1}^\pi \rangle \right],$$

*where $\tilde{V}_{H+1}^\pi \equiv 0 \in \mathbb{R}^S$ and the expectation is over the sampled state trajectory $s_1, \ldots s_H$ drawn from following $\pi$ in the MDP $\overline{M}$.*

*Proof.*

$$\overline{V}_1^\pi(s_1) - \tilde{V}_1^\pi(s_1)$$
$$= \overline{R}_{1, s_1, \pi(s_1)} + \langle \overline{P}_{1, s_1, \pi(s_1)}, \overline{V}_2^\pi \rangle - \tilde{R}_{1, s_1, \pi(s_1)} - \langle \tilde{P}_{1, s_1, \pi(s_1)}, \tilde{V}_2^\pi \rangle$$
$$= \overline{R}_{1, s_1, \pi(s_1)} - \tilde{R}_{1, s_1, \pi(s_1)} + \langle \overline{P}_{1, s_1, \pi(s_1)} - \tilde{P}_{1, s_1, \pi(s_1)}, \tilde{V}_2^\pi \rangle + \langle \overline{P}_{1, s_1, \pi(s_1)}, \overline{V}_2^\pi - \tilde{V}_2^\pi \rangle$$
$$= \overline{R}_{1, s_1, \pi(s_1)} - \tilde{R}_{1, s_1, \pi(s_1)} + \langle \overline{P}_{1, s_1, \pi(s_1)} - \tilde{P}_{1, s_1, \pi(s_1)}, \tilde{V}_2^\pi \rangle + \mathbb{E}_{\pi, \overline{M}} \left[ \overline{V}_2^\pi(s_2) - \tilde{V}_2^\pi(s_2) \right].$$

Expanding this recursion gives the result. $\square$

**Sufficient optimism through randomization.** There is always the risk that, based on noisy observations, an RL algorithm incorrectly forms a low estimate of the value function at some state. This may lead the algorithm to avoid that state, therefore failing to gather the data needed to correct its faulty estimate. To avoid such scenarios, nearly all provably efficient RL exploration algorithms build purposefully optimistic estimates. RLSVI does not do this and instead generates a randomized value

function. The following lemma is key to our analysis. It shows that, except in the rare event when it has grossly mis-estimated the underlying MDP, RLSVI has at least a constant chance of sampling an optimistic value function. Similar results can be proved for Thompson sampling with linear models [1]. Recall $M$ is the unknown true MDP with optimal policy $\pi^*$ and $\overline{M}^k$ is RLSVI's noise perturbed MDP under which $\pi^k$ is an optimal policy.

**Lemma 4.** *Let $\pi^*$ be an optimal policy for the true MDP $M$. If $\hat{M}^k \in \mathcal{M}^k$, then*
$$\mathbb{P}\left(V(\overline{M}^k, \pi^k) \geq V(M, \pi^*) \mid \mathcal{H}_{k-1}\right) \geq \Phi(-1).$$

This result is more easily established through the following lemma, which avoids the need to carefully condition on the history $\mathcal{H}_{k-1}$ at each step. We conclude with the proof of Lemma 4 after.

**Lemma 5.** *Fix any policy $\pi = (\pi_1, \ldots, \pi_H)$ and vector $e \in \mathbb{R}^{HSA}$ with $e(h, s, a) \geq 0$. Consider the MDP $M = (H, \mathcal{S}, \mathcal{A}, P, R, s_1)$ and alternative $\bar{R}$ and $\bar{P}$ obeying the inequality*
$$-\sqrt{e(h, s, a)} \leq \bar{R}_{h,s,a} - R_{h,s,a} + \langle \bar{P}_{h,s,a} - P_{h,s,a}, V_{h+1} \rangle \leq \sqrt{e(h, s, a)}$$
*for every $s \in \mathcal{S}, a \in \mathcal{A}$ and $h \in \{1, \ldots, H\}$. Take $W \in \mathbb{R}^{HSA}$ to be a random vector with independent components where $w(h, s, a) \sim N(0, HSe(h, s, a))$. Let $\bar{V}_{1,W}^\pi$ denote the (random) value function of the policy $\pi$ under the MDP $\bar{M} = (H, \mathcal{S}, \mathcal{A}, \bar{P}, \bar{R} + W)$. Then*
$$\mathbb{P}\left(\bar{V}_{1,W}^\pi(s_1) \geq V_1^\pi(s_1)\right) \geq \Phi(-1).$$

*Proof.* To start, we consider an arbitrary deterministic vector $w \in \mathbb{R}^{HSA}$ (thought of as a possible realization of $W$) and evaluate the gap in value functions $\bar{V}_{1,w}^\pi(s_1) - V_1^\pi(s_1)$. We can re-write this quantity by applying Lemma 3. Let $s = (s_1, \ldots, s_H)$ denote a random sequence of states drawn by simulating the policy $\pi$ in the MDP $\bar{M}$ from the deterministic initial state $s_1$. Set $a_h = \pi(s_h)$ for $h = 1, \ldots, H$. Then
$$\bar{V}_{1,w}^\pi(s_1) - V_1^\pi(s_1)$$
$$= \mathbb{E}\left[\sum_{h=1}^{H} w(h, s_h, \pi_h(s_h)) + \bar{R}_{h,s_h,\pi_h(s_h)} - R_{h,s_h,\pi_h(s_h)} + \langle \bar{P}_{h,s_h,\pi_h(s_h)} - P_{h,s_h,\pi_h(s_h)}, V_{h,w}^\pi \rangle\right]$$
$$\geq H\mathbb{E}\left[\frac{1}{H}\sum_{h=1}^{H}\left(w(h, s_h, \pi_h(s_h)) - \sqrt{e(h, s_h, \pi_h(s_h))}\right)\right]$$

where the expectation is taken over the sequence of sates $s = (s_1, \ldots, s_H)$. Define $d(h, s) = \frac{1}{H}\mathbb{P}(s_h = s)$ for every $h \leq H$ and $s \in \mathcal{S}$. Then the above equation can be written as
$$\frac{1}{H}\left(\bar{V}_{1,w}^\pi(s_1) - V_1^\pi(s_1)\right)$$
$$\geq \sum_{s \in \mathcal{S}, h \leq H} d(h, s)\left(w(h, s, \pi_h(s)) - \sqrt{e(h, s, \pi_h(s))}\right)$$
$$\geq \left(\sum_{s \in \mathcal{S}, h \leq H} d(h, s)w(h, s, \pi_h(s))\right) - \sqrt{HS}\sqrt{\sum_{s \in \mathcal{S}, h \leq H} d(h, s)^2 e(h, s, \pi_h(s))} := X(w)$$

where the second inequality applies Cauchy-Shwartz. Now, since
$$d(h, s)W(h, s, \pi_h(s)) \sim N(0, d(h, s)^2 HSe(h, s, \pi_h(s))),$$
we have
$$X(W) \sim N\left(-\sqrt{HS\sum_{s \in \mathcal{S}, h \leq H} d(h, s)^2 e(h, s, \pi_h(s))}, HS\sum_{s \in \mathcal{S}, h \leq H} d(h, s)^2 e(h, s, \pi_h(s))\right).$$

By standardization, $\mathbb{P}(X(W) \geq 0) = \Phi(-1)$. Therefore, $\mathbb{P}(\bar{V}_{1,W}^\pi(s_1) - V_1^\pi(s_1) \geq 0) \geq \Phi(-1)$. $\square$

*Proof of Lemma 4.* Consider some history $\mathcal{H}_{k-1}$ with $\hat{M}^k \in \mathcal{M}^k$. Recall $\pi^k$ is the policy chosen by RLSVI, which is optimal under the MDP $\overline{M}^k = (H, \mathcal{S}, \mathcal{A}, \hat{P}^k, \hat{R}^k + w^k, s_1)$. Since $\sigma_k^2(h, s, a) = HSe_k(h, s, a)$, applying Lemma 5 conditioned on $\mathcal{H}_{k-1}$ shows that with probability at least $\Phi(-1)$, $V(\overline{M}^k, \pi^*) \geq V(M, \pi^*)$. When this occurs, we always have $V(\overline{M}^k, \pi^k) \geq V(M, \pi^*)$, since by definition $\pi^k$ is optimal under $\overline{M}^k$. $\qquad\qquad\qquad\qquad\qquad\qquad\qquad\qquad\qquad\qquad\quad\square$

**Reduction to bounding online prediction error.** The next lemma shows that the cumulative expected regret of RLSVI is bounded in terms of the total prediction error in estimating the value function of $\pi^k$. The critical feature of the result is it only depends on the algorithm being able to estimate the performance of the policies it actually employs and therefore gathers data about. From here, the regret analysis will follow only concentration arguments. For the purposes of analysis, we let $\tilde{M}^k$ denote an imagined second sample drawn from the same distribution as the perturbed MDP $\overline{M}^k$ under RLSVI. More formally, let $\tilde{M}^k = (H, \mathcal{S}, \mathcal{A}, \hat{P}^k, \hat{R}^k + \tilde{w}^k, s_1)$ where $\tilde{w}^k(h, s, a) \mid \mathcal{H}_{k-1} \sim N(0, \sigma_k^2(h, s, a))$ is independent Gaussian noise. Conditioned on the history, $\tilde{M}^k$ has the same marginal distribution as $\overline{M}^k$, but it is statistically independent of the policy $\pi^k$ selected by RLSVI.

**Lemma 6.** *For an absolute constant $c = \Phi(-1)^{-1} < 6.31$, we have*

$$\text{Regret}(M, K, \text{RLSVI}_\beta) \leq (c+1)\mathbb{E}\left[\sum_{k=1}^{K} |V(\overline{M}^k, \pi^k) - V(M, \pi^k)|\right]$$

$$+ c\mathbb{E}\left[\sum_{k=1}^{K} |V(\tilde{M}^k, \pi^k) - V(M, \pi^k)|\right] + H \underbrace{\sum_{k=1}^{K} \mathbb{P}(\hat{M}^k \notin \mathcal{M}^k)}_{\leq \pi^2/6}.$$

**Online prediction error bounds.** We complete the proof with concentration arguments. Set $\epsilon_R^k(h, s, a) = \hat{R}_{h,s,a}^k - R_{h,s,a} \in \mathbb{R}$ and $\epsilon_P^k(h, s, a) = \hat{P}_{h,s,a}^k - P_{h,s,a} \in \mathbb{R}^S$ to be the error in estimating mean the mean reward and transition vector corresponding to $(h, s, a)$. The next result follows by bounding each term in Lemma 6. This is done by using lemma 3 to expand the terms $V(\overline{M}, \pi^k) - V(M, \pi^k)$ and $V(\overline{M}, \pi^k) - V(\tilde{M}, \pi^k)$. We focus our analysis on bounding $\mathbb{E}\left[\sum_{k=1}^{K} |V(\overline{M}^k, \pi^k) - V(M, \pi^k)|\right]$. The other term can be bounded in an identical manner[2], so we omit this analysis.

**Lemma 7.** *Let $c = \Phi(-1)^{-1} < 6.31$. Then for any $K \in \mathbb{N}$,*

$$\mathbb{E}\left[\sum_{k=1}^{K} |V(\overline{M}^k, \pi^k) - V(M, \pi^k)|\right] \leq \sqrt{\mathbb{E}\sum_{k=1}^{K}\sum_{h=1}^{H-1} \left\|\epsilon_P^k(h, s_h^k, a_h^k)\right\|_1^2} \sqrt{\mathbb{E}\sum_{k=1}^{K}\sum_{h=1}^{H-1} \left\|V_{h+1}^k\right\|_\infty^2}$$

$$+ \mathbb{E}\left[\sum_{k=1}^{K}\sum_{h=1}^{H} |\epsilon_R^k(h, s_h^k, a_h^k)|\right] + \mathbb{E}\left[\sum_{k=1}^{K}\sum_{h=1}^{H} |w^k(h, s_h^k, a_h^k)|\right].$$

The remaining lemmas complete the proof. At each stage, RLSVI adds Gaussian noise with standard deviation no larger than $\tilde{O}(H^{3/2}\sqrt{S})$. Ignoring extremely low probability events, we expect, $\left\|V_{h+1}^k\right\|_\infty \leq \tilde{O}(H^{5/2}\sqrt{S})$ and hence $\sum_{h=1}^{H-1}\left\|V_{h+1}^k\right\|_\infty^2 \leq \tilde{O}(H^6 S)$. The proof of this Lemma makes this precise by applying appropriate maximal inequalities.

**Lemma 8.**

$$\sqrt{\mathbb{E}\sum_{k=1}^{K}\sum_{h=1}^{H-1}\left\|V_{h+1}^k\right\|_\infty^2} = \tilde{O}\left(H^3\sqrt{SK}\right)$$

The next few lemmas are essentially a consequence of analysis in [13], and many subsequent papers. We give proof sketches in the appendix. The main idea is to apply known concentration inequalities to bound $\left\|\epsilon_P^k(h, s_h^k, a_h^k)\right\|_1^2$, $|\epsilon_R^k(h, s_h^k, a_h^k)|$ or $|w^k(h, s_h^k, a_h^k)|$ in terms of either $1/n_k(h, s_h^k, a_h^k)$ or $1/\sqrt{n_k(h, s_h^k, a_h^k)}$. The pigeonhole principle gives $\sum_{k=1}^K \sum_{h=1}^{H-1} 1/n_k(h, s_h^k, a_h^k) = O(\log(SAKH)$ and $\sum_{k=1}^K \sum_{h=1}^{H-1}(1/\sqrt{n_k(h, s_h^k, a_h^k)}) = O(\sqrt{SAKH})$.

**Lemma 9.**
$$\mathbb{E}\left[\sum_{k=1}^K \sum_{h=1}^{H-1} \left\|\epsilon_P^k(h, s_h^k, a_h^k)\right\|_1^2\right] = \tilde{O}\left(S^2 AH\right)$$

**Lemma 10.**
$$\mathbb{E}\left[\sum_{k=1}^K \sum_{h=1}^{H} |\epsilon_R^k(h, s_h^k, a_h^k)|\right] = \tilde{O}\left(\sqrt{SAKH}\right)$$

**Lemma 11.**
$$\mathbb{E}\left[\sum_{k=1}^K \sum_{h=1}^{H} |w^k(h, s_h^k, a_h^k)|\right] = \tilde{O}\left(H^{3/2}S\sqrt{AKH}\right)$$

## 6 Extensions and open directions

This paper gives the first worst-case regret bounds for algorithms that use randomized value functions to drive exploration. That the bounds are polynomial in all parameters indicates that adding noise during value function training generates a sophisticated form of deep exploration that randomizing actions does not [24]. I hope this paper serves as a useful foundation for future analysis, as many questions remain open. One glaring open problem is to study these approaches in problems that require generalization across large state space. Another is to study ensemble approaches [19, 21, 24] that avoid re-estimating the value function in each episode.

There are also clear open questions in the tabular setting. The first, which I am pursuing, is to tighten the dependence on $S$ in the bounds. Another is to tighten the dependence on $H$. I suspect attaining the optimal dependence on $H$ would require adjusting the variances of the noise perturbations in a more adaptive manner. Another question is to extend these proof techniques to handle time-homogeneous MDPs, where there are additional statistical dependencies that would break the current proof. Finally, I believe the proof techniques in this paper could yield high probability bounds on regret. To see this, set $\Delta_k = V(M, \pi^*) - V(M, \pi^k)$ to be the regret incurred in period $k$. Lemma 4 together with the proof of Lemma 7 essentially bounds conditional expected regret $\mathbb{E}[\Delta_k \mid \mathcal{H}_{k-1}]$ with high probability. Since each $\Delta_k$ is bounded, one should be able to apply concentration inequalities to bound the sum of martingale differences $\sum_{k=1}^K (\Delta_k - \mathbb{E}[\Delta_k \mid \mathcal{H}_{k-1}])$ with high probability.

**Acknowledgments.** Much of my understanding of randomized value functions comes from a collaboration with Ian Osband, Ben Van Roy, and Zheng Wen. Mark Sellke and Chao Qin each noticed the same error in the proof of Lemma 6 in the initial draft of this paper. The lemma has now been revised. I am extremely grateful for their careful reading of the paper.

## Footnotes

[1]The precise issue is that, even given a prior over value functions, there is no likelihood function. Given and MDP, there is a well specified likelihood of transitioning from state $s$ to another $s'$, but a value function does not specify a probabilistic data-generating model.

[2]In particular, an analogue of Lemma7 holds where we replace $\overline{M}^k$ with $\tilde{M}^k$, $V_{h+1}^k$ with the value function $\tilde{V}_{h+1}^k$ corresponding to policy $\pi^k$ in the MDP $\tilde{M}^k$, and the Gaussian noise $w^k$ with the fictitious noise terms $\tilde{w}^k$.

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
