[Supplementary Material]

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

*Proof.* The following construction is the standard way concentration inequalities are applied in bandit models and tabular reinforcement learning. See the discussion of what Lattimore and Szepesvári [18] calls a "stack of rewards" model in Subsection 4.6.

For every tuple $z = (h, s, a)$, generate two i.i.d sequences of random variables $r_{z,n} \sim \mathcal{R}_{h,s,a}$ and $s_{z,n} \sim P_{h,s,a}(\cdot)$. Here $r_{(h,s,a),n}$ denotes the reward and $s_{(h,s,a),n}$ denotes the state transition generated from the $n$th time action $a$ is played in state $s$, period $n$. Set

$$Y_{z,n} = r_{z,n} + V^*_{h+1}(s_{z,n}) \qquad n \in \mathbb{N}.$$

These are i.i.d, with $Y_{z,n} \in [0, H]$ since $\|V^*_{h+1}\|_\infty \leq H - 1$, and satisfies

$$\mathbb{E}[Y_{z,n}] = R_{h,s,a} + \langle P_{h,s,a}, V^*_{h+1}\rangle.$$

By Hoeffding's inequality, for any $\delta_n \in (0, 1)$,

$$\mathbb{P}\left(\left|\frac{1}{n}\sum_{i=1}^{n} Y_{(h,s,a),i} - R_{h,s,a} - \langle P_{h,s,a}, V^*_{h+1}\rangle\right| \geq H\sqrt{\frac{\log(2/\delta_n)}{2n}}\right) \leq \delta_n.$$

For $\delta_n = \frac{1}{HSAn^2}$, a union bound over $HSA$ values of $z = (h, s, a)$ and all possible $n$ gives

$$\mathbb{P}\left(\bigcup_{h,s,a,n}\left\{\left|\frac{1}{n}\sum_{i=1}^{n} Y_{(h,s,a),i} - R_{h,s,a} - \langle P_{h,s,a}, V^*_{h+1}\rangle\right| \geq H\sqrt{\frac{\log(2/\delta_n)}{2n}}\right\}\right) \leq \sum_{n=1}^{\infty}\frac{1}{n^2} = \frac{\pi^2}{6}.$$

Now, by definition, if $n_k(h, s, a) = n > 0$, we have

$$\hat{R}^k_{h,s,a} + \langle \hat{P}^k_{h,s,a}, V^*_{h+1}\rangle = \frac{1}{n}\sum_{i=1}^{n} Y_{(h,s,a),i}.$$

Therefore, the above shows

$$\mathbb{P}\left(\exists (k, h, s, a) : n_k(h, s, a) > 0, \left|\hat{R}^k_{h,s,a} - R_{h,s,a} + \langle \hat{P}^k_{h,s,a} - P_{h,s,a}, V^*_{h+1}\rangle\right| \geq H\sqrt{\frac{\log\left(2HSAn_k(h, s, a)\right)}{2n_k(s, h, a)}}\right)$$

is upper bounded by $\pi^2/6$. Note that by definition, when $n_k(h, s, a) > 0$ we have

$$\sqrt{e^k(h, s, a)} \geq H\sqrt{\frac{\log\left(2HSAn_k(h, s, a)\right)}{2n_k(s, h, a)}}$$

and hence this concentration inequality holds with $\sqrt{e^k(h, s, a)}$ on the right hand side. When $n_k(h, s, a) = 0$, we have the trivial bound

$$\left|\hat{R}^k_{h,s,a} - R_{h,s,a} + \langle \hat{P}^k_{h,s,a} - P_{h,s,a}, V^*_{h+1}\rangle\right| = |R_{h,s,a} + \langle P_{h,s,a}, V^*_{h+1}\rangle| \leq H \leq e^k(h, s, a)$$

since we have defined the empirical estimates to satisfy $\hat{R}^k_{h,s,a} = 0$ and $\hat{P}^k_{h,s,a}(\cdot) = 0$ in the case that $h, s, a$ has never been played. $\qquad\square$

## A.2 Proof of Lemma 6

**Lemma 6.** *For an absolute constant $c = \Phi(-1)^{-1} < 6.31$, we have*

$$\mathrm{Regret}(M, K, \mathtt{RLSVI}_\beta) \leq (c + 1)\mathbb{E}\left[\sum_{k=1}^{K} |V(\overline{M}^k, \pi^k) - V(M, \pi^k)|\right]$$

$$+ c\mathbb{E}\left[\sum_{k=1}^{K} |V(\tilde{M}^k, \pi^k) - V(M, \pi^k)|\right] + H\underbrace{\sum_{k=1}^{K}\mathbb{P}(\hat{M}^k \notin \mathcal{M}^k)}_{\leq \pi^2/6}.$$

*Proof.* Recall that $\mathcal{H}_{k-1} = \{(s_h^i, a_h^i, r_h^i) : h = 1, \ldots H, i = 1, \ldots, k-1\}$. So conditioned on $\mathcal{H}_{k-1}$, $\overline{M}^k, \pi^k$ and $\tilde{M}^k$ are random only due to the internal randomness of the RLSVI algorithm. Set $\mathbb{E}_k[\cdot] = \mathbb{E}[\cdot \mid \mathcal{H}_{k-1}]$. Suppose that $\hat{M}^k \in \mathcal{M}^k$. Then

$$\mathbb{P}\left(V(\overline{M}^k, \pi^k) \geq V(M, \pi^*)\Big|\mathcal{H}_{k-1}\right) \geq \Phi(-1). \tag{4}$$

We begin with the regret decomposition:

$$\mathbb{E}_k\left[V(M, \pi^*) - V(M, \pi^k)\right] = \mathbb{E}_k\left[V(M, \pi^*) - V(\overline{M}^k, \pi^k)\right] + \mathbb{E}_k\left[V(\overline{M}^k, \pi^k) - V(M, \pi^k)\right]. \tag{5}$$

We focus on the first term. We show

$$V(M, \pi^*) - \mathbb{E}_k\left[V(\overline{M}^k, \pi^k)\right] \leq c\mathbb{E}_k\left[\left(V(\overline{M}^k, \pi^k) - \mathbb{E}_k\left[V(\overline{M}^k, \pi^k)\right]\right)^+\right]. \tag{6}$$

The inequality is immediate if $V(M, \pi^*) < \mathbb{E}_k\left[V(\overline{M}^k, \pi^k)\right]$. We now show this when $a \equiv V(M, \pi^*) - \mathbb{E}_k\left[V(\overline{M}^k, \pi^k)\right] \geq 0$. Then,

$$\begin{aligned}
\mathbb{E}_k\left[\left(V(\overline{M}^k, \pi^k) - \mathbb{E}_k\left[V(\overline{M}^k, \pi^k)\right]\right)^+\right] &\geq a\mathbb{P}_k\left(V(\overline{M}^k, \pi^k) - \mathbb{E}_k\left[V(\overline{M}^k, \pi^k)\right] \geq a\right) \\
&= \left(V(M, \pi^*) - \mathbb{E}_k\left[V(\overline{M}^k, \pi^k)\right]\right)\mathbb{P}_k\left(V(\overline{M}^k, \pi^k) \geq V(M, \pi^*)\right) \\
&\geq \left(V(M, \pi^*) - \mathbb{E}_k\left[V(\overline{M}^k, \pi^k)\right]\right)\Phi(-1),
\end{aligned}$$

where the first step applies Markov's inequality, the second simply plugs in for $a$, and the third uses Equation 4. Dividing each side by $\Phi(-1)$ gives Equation (6). Hence we have shown

$$\mathbb{E}_k\left[V(M, \pi^*) - V(M, \pi^k)\right] \leq c\mathbb{E}_k\left[\left(V(\overline{M}^k, \pi^k) - \mathbb{E}_k\left[V(\overline{M}^k, \pi^k)\right]\right)^+\right] + \mathbb{E}_k\left[V(\overline{M}^k, \pi^k) - V(M, \pi^k)\right]. \tag{7}$$

We complete our argument by bounding $\mathbb{E}_k\left[\left(V(\overline{M}^k, \pi^k) - \mathbb{E}_k\left[V(\overline{M}^k, \pi^k)\right]\right)^+\right]$. For each fixed (nonrandom) policy $\pi$, define

$$\mu(\pi) \equiv \mathbb{E}_k\left[V(\tilde{M}^k, \pi)\right] = \mathbb{E}_k\left[V(\overline{M}^k, \pi)\right].$$

Notice that $\mu(\pi^k) = \mathbb{E}_k\left[V(\tilde{M}^k, \pi^k) \mid \pi^k\right]$ almost surely. This relies on the fact that $\tilde{M}^k$ and $\pi^k$ are independent conditioned on the history $\mathcal{H}_{k-1}$. In general $\mu(\pi^k) \neq \mathbb{E}_k\left[V(\overline{M}^k, \pi^k) \mid \pi^k\right]$, since $\pi^k$ is the optimal policy under $\overline{M}^k$ and so these two are statistically dependent. Now, for every policy $\pi$

$$\mu(\pi) = \mathbb{E}_k\left[V(\overline{M}^K, \pi)\right] \leq \mathbb{E}_k\left[\sup_{\pi'} V(\overline{M}^K, \pi')\right] = \mathbb{E}_k\left[V(\overline{M}^K, \pi^k)\right].$$

So, $\mu(\pi^k) \leq \mathbb{E}_k\left[V(\overline{M}^K, \pi^k)\right]$ almost surely. Using this, we find

$$\begin{aligned}
\mathbb{E}_k\left[\left(V(\overline{M}^k, \pi^k) - \mathbb{E}_k\left[V(\overline{M}^k, \pi^k)\right]\right)^+\right] &\leq \mathbb{E}_k\left[\left(V(\overline{M}^k, \pi^k) - \mu(\pi^k)\right)^+\right] \\
&\leq \mathbb{E}_k\left[\left|V(\overline{M}^k, \pi^k) - \mu(\pi^k)\right|\right] \\
&= \mathbb{E}_k\left[\left|V(\overline{M}^k, \pi^k) - \mathbb{E}_k\left[V(\tilde{M}^k, \pi^k) \mid \pi^k\right]\right|\right] \\
&= \mathbb{E}_k\left[\left|\mathbb{E}_k\left[V(\overline{M}^k, \pi^k) - V(\tilde{M}^k, \pi^k)\Big|\pi^k, \overline{M}^k\right]\right|\right] \\
&\leq \mathbb{E}_k\left[\mathbb{E}_k\left[\left|V(\overline{M}^k, \pi^k) - V(\tilde{M}^k, \pi^k)\right|\Big|\pi^k, \overline{M}^k\right]\right] \\
&= \mathbb{E}_k\left[\left|V(\overline{M}^k, \pi^k) - V(\tilde{M}^k, \pi^k)\right|\right] \\
&\leq \mathbb{E}_k\left[\left|V(\overline{M}^k, \pi^k) - V(M, \pi^k)\right|\right] + \mathbb{E}_k\left[\left|V(\tilde{M}^k, \pi^k) - V(M, \pi^k)\right|\right].
\end{aligned}$$

Plugging this into (7) shows that, for any history $\mathcal{H}_{k-1}$ with $\hat{M}^k \in \mathcal{M}^k$,

$$\mathbb{E}_k\left[V(M,\pi^*) - V(M,\pi^k)\right] \leq (c+1)\mathbb{E}_k\left[\left|V(\overline{M}^k,\pi^k) - V(M,\pi^k)\right|\right] + c\mathbb{E}_k\left[\left|V(\tilde{M}^k,\pi^k) - V(M,\pi^k)\right|\right].$$

In the unlikely event $\hat{M}^k \notin \mathcal{M}^k$, we have the worst case bound

$$0 \leq V(M,\pi^*) - V(M,\pi^k) \leq H.$$

Combing these two cases and taking expectations gives

$$\mathbb{E}\left[V(M,\pi^*) - V(M,\pi^k)\right] \leq H\mathbb{P}(\hat{M}^k \notin \mathcal{M}^k) + (c+1)\mathbb{E}\left[\left|V(\overline{M}^k,\pi^k) - V(M,\pi^k)\right|\right] + c\mathbb{E}\left[\left|V(\tilde{M}^k,\pi^k) - V(M,\pi^k)\right|\right].$$

Summing over $k$ concludes the proof. $\qquad\square$

### A.3 Proof of Lemma 7

**Lemma 7.** *Let $c = \Phi(-1)^{-1} < 6.31$. Then for any $K \in \mathbb{N}$,*

$$\mathbb{E}\left[\sum_{k=1}^{K}|V(\overline{M}^k,\pi^k) - V(M,\pi^k)|\right] \leq \sqrt{\mathbb{E}\sum_{k=1}^{K}\sum_{h=1}^{H-1}\left\|\epsilon_P^k(h,s_h^k,a_h^k)\right\|_1^2}\sqrt{\mathbb{E}\sum_{k=1}^{K}\sum_{h=1}^{H-1}\left\|V_{h+1}^k\right\|_\infty^2}$$
$$+ \mathbb{E}\left[\sum_{k=1}^{K}\sum_{h=1}^{H}|\epsilon_R^k(h,s_h^k,a_h^k)|\right] + \mathbb{E}\left[\sum_{k=1}^{K}\sum_{h=1}^{H}|w^k(h,s_h^k,a_h^k)|\right].$$

*Proof.* We bound each term in the bound in Lemma 6. By applying Lemma 3 with a choice of $\overline{M} = M$ and $\tilde{M} = \overline{M}^K$, the largest term is bounded, for any $k \in \mathbb{N}$, as

$$\left|V(\overline{M}^k,\pi^k) - V(M,\pi^k)\right|$$
$$= \left|\mathbb{E}\left[\sum_{h=1}^{H}\left(\langle\hat{P}_{h,s_h^k,a_h^k}^k - P_{h,s_h^k,a_h^k}, V_{h+1}^k\rangle\right) + \hat{R}_{h,s_h^k,a_h^k}^k + w^k(h,s_h^k,a_h^k) - R_{h,s_h^k,a_h^k}\,\middle|\,\pi^k,\mathcal{H}_{k-1}\right]\right|$$
$$\leq \mathbb{E}\left[\sum_{h=1}^{H-1}\left\|\epsilon_P^k(h,s_h^k,a_h^k)\right\|_1\left\|V_{h+1}^k\right\|_\infty\,\middle|\,\pi^k,\mathcal{H}_{k-1}\right] + \mathbb{E}\left[\sum_{h=1}^{H}\left(|\epsilon_R^k(h,s_h^k,a_h^k)| + |w^k(h,s_h^k,a_h^k)|\right)\,\middle|\,\pi^k,\mathcal{H}_{k-1}\right]$$

Taking expectations, summing over $k$, and applying Cauchy-Schwartz gives

$$\mathbb{E}\left[\sum_{k=1}^{K}\left|V(\overline{M}^k,\pi^k) - V(M,\pi^k)\right|\right] \leq \sqrt{\mathbb{E}\sum_{k=1}^{K}\sum_{h=1}^{H-1}\left\|\epsilon_P^k(h,s_h^k,a_h^k)\right\|_1^2}\sqrt{\mathbb{E}\sum_{k=1}^{K}\sum_{h=1}^{H-1}\left\|V_{h+1}^k\right\|_\infty^2}$$
$$+ \mathbb{E}\left[\sum_{k=1}^{K}\sum_{h=1}^{H}|\epsilon_R^k(h,s_h^k,a_h^k)|\right] + \mathbb{E}\left[\sum_{k=1}^{K}\sum_{h=1}^{H}|w^k(h,s_h^k,a_h^k)|\right].$$

$\qquad\square$

### A.4 Proof of Lemma 8

The proof relies on the following maximal inequality.

**Lemma 12** (Example 2.7 From [7])**.** *If $X_1,\dots,X_n$ are i.i.d. random variables following a $\chi_1^2$ distribution, then*

$$\mathbb{E}\left[\max_{i\leq n}X_i\right] \leq 1 + \sqrt{2\log(n)} + 2\log(n).$$

Let us now recall Lemma 8.

**Lemma 8.**

$$\sqrt{\mathbb{E}\sum_{k=1}^{K}\sum_{h=1}^{H-1}\left\|V_{h+1}^k\right\|_\infty^2} = \tilde{O}\left(H^3\sqrt{SK}\right)$$

*Proof.* We have

$$\sqrt{\mathbb{E}\sum_{k=1}^{K}\sum_{h=1}^{H-1}\left\|V_{h+1}^{k}\right\|_{\infty}^{2}} \leq \sqrt{HK\mathbb{E}\left[\max_{k\leq K, h\leq H}\left\|V_{h+1}^{k}\right\|_{\infty}^{2}\right]}$$

Now

$$V_{h+1}^{k}(s') \leq (H-h-1)(1+\max_{h,s,a} w^{k}(h,s,a))$$

$$V_{h+1}^{k}(s') \geq (H-h-1)(\min_{h,s,a} w^{k}(h,s,a)).$$

Together this gives that for all $k \leq K$ and $h \in \{1, \ldots, H-1\}$

$$\left\|V_{h+1}^{k}\right\|_{\infty} \leq H\left(1+\max_{k\leq K,h,s,a}|w^{k}(h,s,a)|\right)^{2} \leq 4H^{2}+4H^{2}\left(\max_{k\leq K,h,s,a}|w^{k}(h,s,a)|^{2}\right).$$

We have $w^{k}(h,s,a) = \sigma_{k}(h,s,a)\xi_{h,s,a}^{k}$ where the $\xi_{h,s,a}^{k} \sim N(0,1)$ are drawn i.i.d across $h,s,a$. Set $X_{h,s,a}^{k} = (\xi_{h,s,a}^{k})^{2}$, each of which follows a chi-squared distribution with 1 degree of freedom. Then,

$$
\begin{aligned}
\mathbb{E}\left[\max_{k\leq K,h,s,a}|w^{k}(h,s,a)|^{2}\right] &\leq \left(\max_{k\leq K,h,s,a,}\sigma_{k}^{2}(h,s,a)\right)\mathbb{E}\left[\max_{k\leq K,h,s,a}|\xi_{h,s,a}^{k}|^{2}\right] \\
&= \left(\max_{k\leq K,h,s,a,}\sigma_{k}^{2}(h,s,a)\right)\mathbb{E}\left[\max_{k\leq K,h,s,a}X_{h,s,a}^{k}\right] \\
&\leq \left(SH^{3}\log(2SAHK)\right)\mathbb{E}\left[\max_{k\leq K,h,s,a}X_{h,s,a}^{k}\right] \\
&\leq \left(SH^{3}\log(2SAHK)\right)\left(1+\sqrt{2\log(SAHK)}+2\log(SAHK)\right) \\
&\leq O\left(SH^{3}\log(2SAHK)^{2}\right).
\end{aligned}
$$

This gives us

$$\sqrt{KH\mathbb{E}\left[\max_{k\leq K,h\leq H}\left\|V_{h+1}^{k}\right\|_{\infty}^{2}\right]} = \tilde{O}\left(\sqrt{KH\cdot H^{2}\cdot SH^{3}}\right) = \tilde{O}\left(H^{3}\sqrt{SK}\right).$$

$\square$

### A.5 Proof sketch of Lemma 9

This result relies on an inequality by Weissman et al. [30], which we now restate.

**Lemma 13.** *[L1 deviation bound] If $p$ is a probability distribution over $\mathcal{S} = \{1, \ldots S\}$ and $\hat{p}$ is the empirical distribution constructed from $n$ i.i.d draws from $p$, then for any $\epsilon > 0$,*

$$\mathbb{P}\left(\|\hat{p}-p\|_{1} \geq \epsilon\right) \leq (2^{S}-2)\exp\left(-\frac{n\epsilon^{2}}{2}\right)$$

**Lemma 9.**

$$\mathbb{E}\left[\sum_{k=1}^{K}\sum_{h=1}^{H-1}\left\|\epsilon_{P}^{k}(h,s_{h}^{k},a_{h}^{k})\right\|_{1}^{2}\right] = \tilde{O}\left(S^{2}AH\right)$$

*Proof sketch.* By picking an appropriate $\epsilon$ in Lemma 13 as in [13, Appendix C.1], together with a union bound over all $HSA$ possible values for the tuple $(h,s,a)$, there exists a numerical constant $c$ such that

$$\mathbb{P}\left(\bigcup_{s,a,h,k\leq K}\left\{\left\|\hat{P}_{h,s,a}^{k}-P_{h,s,a}\right\|_{1} \geq c\sqrt{\frac{S\log(1+HSAK)}{n_{k}(h,s,a)}+1}\right\}\right) \leq \frac{1}{KH}. \quad (8)$$

Set $\beta_k(h, s, a) = \frac{S\ell}{n_k(h,s,a)}$ where $\ell = c^2 \log(1 + HSAK)$ denotes a logarithmic factor. Recall the definition $\epsilon_P^k(h, s, a) \equiv \hat{P}_{h,s,a}^k - P_{h,s,a}$. Let $B$ be the "bad event" that $\|\epsilon_P^k(h, s, a)\|_1^2 \geq \beta_k(h, s, a)$ for some $(h, s, a)$ and $k \leq K$. Since $\|\epsilon_P^k(h, s, a)\|_1 \leq 2$ always, we have

$$\mathbb{E} \sum_{k=1}^{K} \sum_{h=1}^{H-1} \|\epsilon_P^k(h, s_h^k, a_h^k)\|_1^2 \mathbf{1}(B) \leq 4 \tag{9}$$

On the other hand, assuming $B^c$ we have the bound

$$\sum_{k=1}^{K} \sum_{h=1}^{H-1} \|\epsilon_P^k(h, s_h^k, a_h^k)\|_1^2 \leq \sum_{k=1}^{K} \sum_{h=1}^{H-1} \beta_k(h, s_h^k, a_h^k) = S\ell \sum_{k=1}^{K} \sum_{h=1}^{H-1} \frac{1}{n_k(h, s_h^k, a_h^k) + 1}$$

$$\leq \sum_{h,s,a} \sum_{n=0}^{n_K(h,s,a)} \frac{1}{n+1}$$

$$= O\left(HSA \log(K)\right).$$

$\square$

## A.6  Proof sketch of Lemma 10

**Lemma 10.**

$$\mathbb{E}\left[\sum_{k=1}^{K} \sum_{h=1}^{H} |\epsilon_R^k(h, s_h^k, a_h^k)|\right] = \tilde{O}\left(\sqrt{SAKH}\right)$$

*Proof sketch.* The proof is similar to Lemma 9. By Hoeffding's inequality together with a union bound, we can ensure that $|\epsilon_R^k(h, s, a)| \leq c\sqrt{\frac{\log(1+HSAK)}{n_k(h,s,a)+1}}$ for all $k \leq K$ and all tuples $(h, s, a)$ except on some bad event that, as in (9), contributes at most a constant to the bound. Now the result follows from using the pigeonhole principle to conclude

$$\sum_{k=1}^{K} \sum_{h=1}^{H} \frac{1}{\sqrt{n_k(h, s_h^k, a_h^k)}} = O\left(\sqrt{HSAK}\right).$$

This kind of bound bound is standard in the RL and bandit literature. See [20, Appendix A] for one proof. $\square$

## A.7  Proof sketch of Lemma 11

**Lemma 11.**

$$\mathbb{E}\left[\sum_{k=1}^{K} \sum_{h=1}^{H} |w^k(h, s_h^k, a_h^k)|\right] = \tilde{O}\left(H^{3/2} S\sqrt{AKH}\right)$$

*Proof.* Recall $\sigma_k(h, s, a) = \sqrt{\frac{\beta}{n_k(h,s,a)+1}}$ where $\beta_k = \tilde{O}(SH^3)$. Write $w^k(h, s, a) = \sigma^k(h, s, a)\xi_k(h, s, a)$ where $\xi_k(h, s, a) \sim N(0, 1)$ and the array of random variable $\{\xi_k(h, s, a) : 1 \leq k \leq K, 1 \leq h \leq H, a \in \mathcal{A}, s \in \mathcal{S}\}$ is drawn independently. By Holder's inequality,

$$\mathbb{E} \sum_{k=1}^{K} \sum_{h=1}^{H} |w_k(h, s_h^k, a_h^k)| \leq \mathbb{E}\left(\max_{k \leq K, h, s, a} |\xi_k(h, s, a)|\right) \mathbb{E} \sum_{k=1}^{K} \sum_{h=1}^{H} \sigma_k(h, s_h^k, a_h^k)$$

The (sub) Gaussian maximal inequality gives

$$\mathbb{E}\left(\max_{k \leq K, h, s, a} |\xi_k(h, s, a)|\right) = O\left(\sqrt{\log(HSAK)}\right).$$

To simplify the next expression, note that $\beta_k \leq \beta_K$. On any sample path, by the same argument as in Lemma 10, we have

$$\sum_{k=1}^{K} \sum_{h=1}^{H} \sigma_k(h, s_h^k, a_h^k) \leq \beta_K \sum_{k=1}^{K} \sum_{h=1}^{H} \sqrt{\frac{1}{n_k(h, s_h^k, a_h^k) + 1}} = O\left(\beta_K \sqrt{HSAK}\right).$$

$\square$