[Reviews · NeurIPS 2019]

Reviewer 1



Post author response: I thank the author(s) for their response and commenting on my discussion points. As those would need additional work, I for now keep my original score: this is a solid paper. ---------------------- Clarity: The paper is very well written and generally easy to understand, given the technical nature of the contribution. While the proof for Lemma 4 & 5 is described very well in the main text, it would be helpful to have a short explanation how this is used to achieve Lemma 6. If necessary, I suggest to drop the proof of Lemma 3 from the main text as this result is standard. Quality: I have verified the proof in the main text and individual lemmas in the appendix. All results that I checked appear to be correct. The authors do discuss the main limitations of their analysis but it would be nice to discuss the following points at least briefly: - The authors chose the setting with time-dependent dynamics which is a little less common than the default setting where dynamics and rewards are identically distributed across time steps within the episode. Would the current analysis be able to go through even if samples are shared across time steps? I suspect it would be difficult to achieve a tighter bound in H (e.g. due to Lemma 8) but I wonder whether the same bound can be proved. - How does this setting for \beta affect the empirical performance of the algorithm. To what degree does it make the algorithm conservative compared to empirically tuned parameters? - This is an expected regret bound (given any fixed true MDP) but is there any indication that a similar analysis would not yield a high-probability regret bound or PAC-style bound (as in [11] after adjusting k --> n_k(h,s,a) in beta)? Originality: This is the first worst-case analysis for RLSVI and even though it might not yield novel insights beyond the analysis itself, is an important technical contribution. The proof relies to large parts on existing strategies (which is not surprising) but contains novel technical insights such as lemma 5. The paper does discuss the relevant links to existing literature but it would have been helpful to give a more direct comparison of the result here to the lower bound and upper bounds of other algorithms (e.g. OFU based methods) in this setting. Significance: I think this is a solid technical contribution to the field and even though it is not surprising or particularly tight, it is absolutely non-trivial and still an important result. As mentioned in the introduction of the paper itself, worst-case analyses of randomized algorithms in the RL setting are far from trivial and so this relatively simple analysis has a good chance of being used as a template and improved upon in future work. Minor: - Line 361: hat M_k \in Mcal_k should be \notin - Line 264 missing closing )

Reviewer 2



Originality: while the principle is certainly common in the literature, this is the first paper to demonstrate frequentist regret guarantees for perturbation induced exploration. Proof techniques do not appear to be terribly different than the prior arm, with the key differences appearing in Lemmas 4-5. Quality: The work appears generally correct, and the proofs are readable and succinct. Clarity: the proofs appear generally correct however, I believe that the third inequality beneath line 360 in the appendix could be explained more thoroughly, since justifying this statement crucially relies on conditioning on H_{k-1}, which is not indicated in the notation. Moreover, I think in the description of the algorithm, it should be stressed that the data augmentation is solely for planning, and not for estimation of transition probabilities. Significance: I think the significance of the paper lies in the argument that optimism is not a realistic framework for more complicated RL settings, and that randomized data augmentation gives a more plausible approach which is still amenable to theoretical analysis. However, I do not know if the fact that randomized value functions do in fact ensure frequentist regret guarantees is terribly surprising. In sum, my inclination to accept the paper is not so much for its technical novelty, as for bringing to light the fact that there are other algorithm design paradigms for RL that may generalize better than optimism in more complex settings, but which are nevertheless theoretically sound.

Reviewer 3



RLSVI belongs to a family of algorithms that can be used with function approximation to encourage exploration. While the method enjoys some theoretical guarantees in the tabular setting, previous analyses provide Bayesian regret bound, and this paper addresses the frequentist (worst-case) regret. The writing is clear and the text provides interesting discussions and insights around the problem. In general I like the paper, and I have several clarification questions for the authors: 1. Regarding worst-case regret bounds for Thompson-sampling-like methods, it reminds me of the work of Agrawal and Jia [3]. If you look at their algorithm, it's actually two-phased: they use optimism when the sample size is small, and only switch to legit Thompson sampling in the large sample regime. What's interesting to me about the current paper is that there are no such two-phased structure, which looks a little bit too good and I will explain why: RLSVI is similar and closely related to bootstrapping, and we know that bootstrapping only works when the data somewhat resemble the true distribution. This requirement can be difficult to satisfy in MDPs: unlike bandits where you can just sample each action a small number of times to achieve full coverage, in MDPs we often find ourselves not having any data at all from certain states in the middle of exploration. Now I guess this is circumvented by choosing \beta_k to be large enough in the beginning---in fact I was expecting \beta_k to start with O(H) value, the magnitude of the reward function, only to find that it starts with O(H^3)... but this does not fully answer my question: high noise would simply induce uniformly random exploration, which is for sure inefficient (e.g., in combination locks). Can you give some insights on this matter? 2. Algorithm 1 is very nice in the sense that one can almost directly use it in the function approximation setting (e.g., there is no mention of the visitation counters), except for Line 6 that samples the \tilde{Q} from a tabular prior. Some discussions about what's the counterpart of this step in the function approximation setting can be interesting. ----------------Update after rebuttal---------------- Thanks for the response, and it makes a lot of sense. I keep my original recommendation.

[Author Response · NeurIPS 2019]

Thank you to all reviewers! The response to reviewer 2 may be most important to how the paper is interpreted:

**Reviewer 2:** *while the principle is certainly common in the literature, this is the first paper to demonstrate frequentist*
*regret guarantees for perturbation induced exploration. Proof techniques do not appear to be terribly different than the*
*prior arm, with the key differences appearing in Lemmas 4-5.. . . . . . However, I do not know if the fact that randomized*
*value functions do in fact ensure frequentist regret guarantees is terribly surprising.*

Managing to give such a clean, seemingly straightforward, frequentist analysis of RLSVI seems to be a major
contribution over prior art. The journal paper Osband et al. [2017] develops a theory of recursive stochastic-dominance
relations to study the algorithm, hence requiring very different techniques than the rest of the RL literature. The paper on
frequentist analysis of posterior sampling by Agrawal and Jia [2017] builds on those stochastic-dominance techniques,
is immensely technical, requires modifying Thompson sampling to get the proof to work, and contains a critical flaw in
the proof currently posted online. I've worked very hard to uncover a new proof that hopefully makes it easy for future
researchers to transfer results known for optimistic algorithms over to randomized value function approaches.

On whether the results are surprising: Strong theory sometimes takes years to develop and in the meantime people
can start to get used to the main ideas. This paper tries to provide some backing for the claim that "Training a
value function estimation scheme on noise-perturbed data generates a highly sophisticated form of exploration that is
fundamentally quite different from what is generated by employing stochastic policies." It has been just three years
since the first paper making such a claim was published [Osband et al., 2016]. To my understanding, this claim was
often met with skepticism, especially because it lacked a frequentist regret bound to back it up. Such a bound has
been elusive since then. Things can seem quite clear in hindsight, but I think the claims in this paper would have been
shocking 7 years ago (before any analysis of Thompson sampling in bandits even exited.) That's noteworthy, since
we're studying extremely old questions in sequential decision making.

**Responses to Reviewer 1:** [Paraphrasing] *(1) Would a similar analysis yield a high-probability regret bound? . . . (2)*
*While the proof for Lemma 4 & 5 is described very well in the main text, it would be helpful to have a short explanation*
*how this is used to achieve Lemma 6.. . . (3) How does this setting for $\beta$ affect the empirical performance of the algorithm.*
*. . . (4) The authors chose the setting with time-dependent dynamics which is a little less common than the default setting*
*where dynamics and rewards are identically distributed across time steps within the episode. . .*

(1) Thanks so much! I now believe the high probability bounds work out. Effectively, the techniques in this paper bound,
with high probability, the conditional expected regret $\mathbb{E}[V(M, \pi^*) - V(M, \pi^k) \mid \mathcal{H}^k] \leq B_k$ by some simple terms $B_k$
whose sum we know how to control. Rather than take an expectation, the Azuma-Hoeffdin'g inequality should bound
the sum of martingale differences: $\sum_{k=1}^{K} \left( \left(V(M, \pi^*) - V(M, \pi^k)\right) - \mathbb{E}[V(M, \pi^*) - V(M, \pi^k) \mid \mathcal{H}_{k-1}] \right)$ The main
challenge seems to be that $B_k$ involves terms that are not uniformly bounded, since we add Gaussian noise. The
standard ways in which researchers bounds things like $B_1 + \cdots + B_K$ could get unusually messy as a result.

(2-4) In the revision, I'll explain more about Lemma 6 and clearly highlight open questions regrading points (3) and (4).
I think setting $\beta$ too large is similar to setting overly large optimism bonuses for optimistic algorithms. For practical
applications, I suspect the noise variances one adds should also be adaptive to the data. We're essentially (recursively)
applying linear-regression to minimize the Bellman residuals. One can write down variants of RLSVI that calibrate the
noise they add to the variance of observed residuals, but I'd like to do careful empirical evaluation. I'll try to work
through the time-homogeneous analysis. The challenge is that mis-estimation of a single state could lead to error in
every Bellman update. One needs to be careful to avoid introducing even more factors of $H$.

**Responses to Reviewer 3:** *In response to the comments on bootstrapping and the correct form of prior randomness .*

Thanks, this really gets to the crux of what makes boostrap-like methods for exploration so different from the treatment
statistics books. You're right that in initial periods, RLSVI is really no better than uniform exploration, though perhaps
the same comment applies to optimistic algorithms. But my understanding is that, no matter what, any algorithm will
visit certain states/actions many times. These become well understood and the variance of the injected noise begins to
vanish at those states/actions. Some other parts of the state/action space are still poorly understood. Because we add
lots of noise to the estimated rewards at those states, there is a significant chance they appear to be even better than they
really are in any given episode, in which case the algorithm will deftly navigate through the well understood part of
the state space trying to reach them. Once the algorithm starts actively trying to reach poorly understood states, it's
performing the kind of multi-period exploration that's essential for efficient RL.

To inject prior randomness, there is a natural counterpart (to sampling $Q$) for linear models, where you regularize the
parameter vector to a prior sample. This proecdure actually corresponds to Algorithm 1 if specialize it to the tabular
case (think $\theta = Q$). Things are more subtle with neural networks but [Osband et al., 2018] offers one approach. You're
right, that algorithm otherwise applies directly to settings with function approximation.

[Meta-Review · NeurIPS 2019]

The paper gives a frequentist regret bound for the RLSVI algorithm. While the bound is not minimax optimal (and potentially can be improved), this is the first frequentist guarantee for this algorithm and the proof contains some new technical insights, which may be useful in future work. Further the result demonstrates that other algorithmic strategies/paradigms (besides say optimism) may yield provably sample-efficient RL methods. Thanks for notifying us about a bug that you found in the proof! I discussed this with the reviewers and we all decided it was not a deal breaker for us. Anyway the regret bound here is suboptimal, so the fact that the actual result is slightly worse is fairly insignificant.